# Exploring the Effects of Various Two-Dimensional Supporting Materials on the Water Electrolysis of Co-Mo Sulfide/Oxide Heterostructure

**DOI:** 10.3390/nano13172463

**Published:** 2023-08-31

**Authors:** Ngoc-Diem Huynh, Won Mook Choi, Seung Hyun Hur

**Affiliations:** School of Chemical Engineering, University of Ulsan, Daehak-ro 93, Nam-gu, Ulsan 44610, Republic of Korea; diemhuynh0908@gmail.com

**Keywords:** oxygen evolution reaction, water splitting, cobalt molybdenum sulfide, cobalt molybdenum oxide, reduced graphene oxide

## Abstract

In this study, various two-dimensional (2D) materials were used as supporting materials for the bimetallic Co and Mo sulfide/oxide (CMSO) heterostructure. The water electrolysis activity of CMSO supported on reduced graphene oxide (rGO), graphite carbon nitride (gC_3_N_4_), and siloxene (SiSh) was better than that of pristine CMSO. In particular, rGO-supported CMSO (CMSO@rGO) exhibited a large surface area and a low interface charge-transfer resistance, leading to a low overpotential and a Tafel slope of 259 mV (10 mA/cm^2^) and 85 mV/dec, respectively, with excellent long-term stability over 40 h of continuous operation in the oxygen evolution reaction.

## 1. Introduction

Hydrogen produced by electrolysis is one of the sustainable and promising energies to alter the energy from fossil fuels because of several factors, including the abundance of water as a feedstock, it being free of carbon dioxide emissions, and its wide range of applications [1]. However, the efficiency and utilization of water electrolysis are hindered by the low kinetics of the oxygen evolution reaction (OER) [2], which is the half-reaction of the water-splitting reaction.

The heterostructure material, which is composed of two or more components, possesses the synergistic effect of all components to overcome the disadvantages of individual ones. Additionally, the contact of the crystal components might change the electronic structure and strain the material, rendering it appropriate for OER [3,4,5]. As a result, various transition metals have been integrated in heterogeneous manners [6,7,8] to replace the scarce and expensive benchmark precious-metal-based electrocatalysts such as RuO_2_ or IrO_2_ in the OER process [9]. Among the transition metals, cobalt (Co) and molybdenum (Mo) are considered to be the most promising candidates because of their earth abundance and cost-effectiveness, as well as the excellent redox behavior of Co and high electrical conductivity of Mo [10,11,12,13]. 

Recently, several studies revealed that the introduction of a sulfur anion into Co is favorable for driving water oxidation. Wang et al. confirmed that the moderate replacement of oxygen with sulfur could modify the electronic structure of the composite to achieve optimal intrinsic OER activity [14]. Fei et al. reported that the co-substitution of Fe and S in CoMoO_4_ increased the charge-transfer ability and decreased the energy barrier of the rate-determining step during OER [15]. Hu et al. confirmed that compared to a pure oxide surface, a lattice oxygen–sulfur co-existing shell surface of (NiCo)O_x_S_1.33−x_ lowered the applied potential for surface reconstruction [16]. 

Employing a two-dimensional (2D) material as a supporting material for electrocatalysts can be an efficient strategy to increase the number of active sites and improve long-term stability [17]. Reduced graphene oxide (rGO) has been widely used as a supporting material owing to its high specific surface area, high conductivity, and excellent mechanical strength [18]. As another carbon-based 2D material, graphitic carbon nitride (gC_3_N_4_) can also be used as a support because of its facile availability, simple production route, cost-effectiveness, and excellent chemical and thermal robustness [19,20]. Additionally, as a hexagonal 2D material composed of six-membered rings of silicon separated from each other by Si–O–Si bridges, siloxene (SiSh) also exhibits excellent properties when used as a support [21,22], owing to the high specific surface area and the presence of hydroxyl groups on the siloxene sheet [23].

Therefore, the heterogeneous structure of an oxide–sulfide composite of CoMoO_4_/CoS/MoS_2_ (CMSO) combined with the 2D material (CMSO@2D) is thought to be a good candidate out of the high-performance anode materials for the water-splitting reaction. This study aimed to explore the potential of 2D materials, including rGO, gC_3_N_4_, and SiSh, as supporting materials for CMSO to enhance the electrochemical activity and stability during the OER process. By conducting instrumental analysis and electrochemical characterization, rGO was found to be the best support for CMSO, and CMSO@rGO exhibited a low OER overpotential and a Tafel slope of 259 mV (10 mA/cm^2^) and 85 mV/dec, respectively, which were comparable to those of RuO_2_. In addition, a clear current drop was not observed even after 40 h of continuous operation.

## 2. Experimental Section

### 2.1. Synthesis of Various 2D Materials

Graphene oxide was prepared by using the modified Hummer’s method, as reported previously [24], and subsequently reduced using hydrazine monohydrate (N_2_H_4_·H_2_O) to obtain rGO nanosheets. SiSh was synthesized using the procedure described previously [22]. To prepare gC_3_N_4_, 10 g of melamine was added into a porcelain crucible and heated at 600 °C for 4 h. Then, the resulting yellow powder was mixed with 100 mL of deionized (DI) water and subjected to sonication. The final product was obtained after centrifugation and subsequent drying overnight at 100 °C in air.

### 2.2. Synthesis of CMSO@2D Materials

The CMSO@2D materials (viz. rGO, gC_3_N_4_, and SiSh) were fabricated using a two-step method of solvothermal synthesis and vacuum annealing, respectively. In the solvothermal step, thioacetamide (TAA) was added as a sulfur source of CMSO. Co-glycerate and ammonium molybdate tetrahydrate ((NH_4_)_6_Mo_7_O_24_·4H_2_O) were used as the precursors for Co^2+^ and Mo^6+^, respectively, which reacted with TAA to form Co/Mo sulfide. The used 2D material served as a support and template for the anchoring sites of Co/Mo sulfide. During the vacuum annealing process, the materials were partially oxidized, resulting in the formation of CoMoO_4_/CoS/MoS_2_@2D (CMSO@2D, Figure 1). 

Typically, mixture A was prepared by dissolving 20 mg of the Co-glycerate precursor (following the procedure in the Appendix A), 10 mg of (NH_4_)_6_Mo_7_O_24_·4H_2_O, and 50 mg of TAA in 20 mL of ethanol. Simultaneously, mixture B was formed by sonicating 20 mg of gC_3_N_4_ in 20 mL of ethanol. Then, mixture A was slowly added to mixture B and stirred for 1 h. The resulting mixture was transferred into a Teflon-lined autoclave and heated, and the temperature was maintained at 200 °C for 6 h. After centrifugation and washing with ethanol and DI water, the product was dried in a vacuum oven at 60 °C for 12 h. Subsequently, the product was annealed at 500 °C under vacuum for 2 h. A similar procedure was followed for the reactions of the other 2D materials, with gC_3_N_4_ substituted by SiSh and rGO. In the case of rGO, 0.1 mL of N_2_H_4_·H_2_O was added to mixture B.

### 2.3. Synthesis of CMSO 

CMSO spheres were synthesized using a process similar to that of CMSO@2D except for the addition of the 2D materials.

## 3. Result and Discussion

### 3.1. Characterization of As-Prepared Materials

The crystal structure and phase composition of all materials were analyzed via X-ray diffraction (XRD), as shown in Figure 1a. The XRD patterns of CMSO@rGO, CMSO@gC_3_N_4_, and CMSO@SiSh were similar to that of CMSO, indicating that a structural change did not occur during the deposition of CMSO on 2D materials and that all samples exhibited distinct MoS_2_, CoMoO_4_, and CoS phases. The diffraction peaks at 14.4° and 29.2° corresponded to the (002) and (004) planes of MoS_2_, respectively (JCPDS No. 037-1492) [25]. The diffraction peaks at 26.5°, 36.5°, 42.4°, 53.5°, 61.5°, 73.7°, and 77.5° were characteristic of the (220), (400), (123), (333), (061), (622), and (350) planes of CoMoO_4_ (JCPDS No. 04-017-6377), respectively [26]. The peaks observed at 34.4° and 47.1° were ascribed to the (101) and (102) planes of hexagonal CoS, respectively (JCPDS No. 65–3418) [27]. 

Additionally, in the XRD pattern of CMSO@SiSh, the characteristic peaks of SiSh were observed at 14.1°, 28.5°, and 56.1°, corresponding to the (001), (111), and (311) planes of siloxene, respectively. The peak located at 27.5° in the XRD pattern of CMSO@gC_3_N_4_ was attributed to gC_3_N_4_ (JCPDS No. 87-1526). Because the specific peak of rGO at 26.8° (JCPDS No. 89-8487) was overlapped with those of MoS_2_ at 28.5° and CoMoO_4_ at 26.5°, distinguishing the rGO-related peaks in the XRD pattern of CMSO@rGO was difficult. The Raman spectra of the prepared materials are shown in Figure 1b. In the Raman spectrum of CMSO, characteristic peaks for the bonding vibrations of Co–O–Mo at 808, 865, and 925 cm^−1^; MoO_4_ at 328 and 352 cm^−1^ [28]; CoS at 511 and 676 cm^−1^ [27]; and MoS_2_ at 280 cm^−1^ were observed [29]. In the Raman spectrum of CMSO@rGO, two additional peaks were observed at 1351 cm^−1^ and 1596 cm^−1^. These peaks were assigned to the D band and G band of rGO, respectively. The integrated area ratio of the D and G bands of pristine rGO (Appendix A) was 1.27, while that of CMSO@rGO increased to 1.46, indicating that after the anchoring of CMSO, the defect density of the rGO surface increased. In the Raman spectrum of CMSO@gC_3_N_4_, a broad peak was observed at approximately 1600 cm^−1^, which was similar to that of bulk gC_3_N_4_ (Appendix A). In the Raman spectrum of CMSO@SiSh, an intense peak at 513 cm^−1^ corresponding to the Si–Si vibration of SiSh was observed, which was same as that of SiSh shown in Appendix A.

Figure 2 shows the Brunauer–Emmett–Teller (BET) analysis of all materials. This result revealed that the material exhibited a type IV isotherm according to the IUPAC classification [30], indicative of the presence of a mesoporous structure with a pore size ranging from 2 to 50 nm. Notably, CMSO@rGO exhibited a significantly higher nitrogen adsorption amount, leading to the highest surface area among all of the materials. According to Appendix A and Table 1, the average pore radii of CMSO, CMSO@rGO, CMSO@gC_3_N_4_, and CMSO@SiSh were approximately distributed at 3.9, 1.6, 1.6, and 2.0 nm, respectively, corresponding to the pore volumes of 0.320, 1.154, 0.463, and 0.398 cm^3^/g.

The specific surface area increased in the order of CMSO (116 m^2^/g) < CMSO@gC_3_N_4_ (156 m^2^/g) < CMSO@SiSh (201 m^2^/g) < CMSO@rGO (1392 m^2^/g), indicating that the modification of CMSO with 2D materials led to an increase in the specific surface area. Notably, the introduction of rGO substantially increased the specific surface area of the composite.

To investigate the morphology of the as-prepared materials, field-emission scanning electron microscopy (FESEM) (Hitachi High-Tech Corporation, SU7000, Tokyo, Japan) was conducted. The FESEM images are shown in Figure 3. CMSO exhibited highly agglomerated nanosphere particles, resulting in the formation of large clusters (Figure 3a). In contrast, when CMSO was supported on 2D materials, especially rGO (Figure 3b) and gC_3_N_4_ (Figure 3c), the interparticle voids were increased, which could provide additional pathways and spaces for the electrolytic ions to access the active sites. CMSO on SiSh (Figure 3d) exhibited a non-uniform morphology and a high degree of agglomeration, which could be attributed to a low number of functional groups that could anchor CMSO nanoparticles. The elemental mapping of all materials is shown in Appendix A. The constitutional elements exhibited a uniform distribution.

The elemental electronic states were investigated using X-ray photoelectron spectroscopy (XPS). In the deconvoluted Mo 3d spectra of CMSO (Figure 4a), dominant peaks observed at 235.4 and 232.3 eV corresponded to 3d_3/2_ and 3d_5/2_ of Mo^6+^, respectively, and those at 234.7 and 231.2 eV corresponded to 3d_3/2_ and 3d_5/2_ of Mo^4+^, respectively [31]. The weak peak located at approximately 228 eV was associated with S 2s [32]. After the addition of 2D materials, these peaks were positively shifted relative to bare CMSO. In the deconvoluted Co 2p spectrum of CMSO (Figure 4b), two peaks located at 780.9 and 796.3 eV, accompanied by two satellite peaks indicated by asterisks, were attributed to Co 2p_3/2_ and Co 2p_1/2_ of Co^2+^, respectively [33]. An additional peak at 779.8 eV was attributed to Co–S bonding [27]. Interestingly, the position of Co 2p peaks in CMSO@rGO, CMSO@gC_3_N_4_, and CMSO@SiSh shifted to higher binding energies compared with that of pristine CMSO, indicative of the loss of electrons in Co [34]. The deconvoluted S 2p spectrum (Figure 4c) revealed four peaks. The peaks observed at 162.6 and 161.3 eV corresponded to S^2−^ of Co–S and Mo–S, respectively [35,36]. Two additional peaks at 169 and 167.7 eV corresponded to oxidized sulfur [32]. The high-resolution O 1s XPS spectra (Figure 4d) showed a major peak at 530.2 eV, corresponding to O in CoMoO_4_ [37]. After modification with 2D materials, the S 2p peaks exhibited a negative shift, and the O 1s peaks exhibited a positive shift, indicative of the electronic interaction between CMSO and 2D materials [34]. Such charge transfer between CMSO and supporting materials could induce adjustments in energy-band alignment, which might thermodynamically facilitate the OER process [38].

### 3.2. Electrocatalytic Activity of As-Prepared Materials

To evaluate the effect of 2D supporting materials on the activity of CMSO in the OER process, the electrocatalytic properties of CMSO, CMSO@rGO, CMSO@gC_3_N_4_, and CMSO@SiSh were investigated and compared with those of RuO_2_, which was widely recognized as a benchmark material for OER. The OER overpotentials of 2D material-supported CMSO such as CMSO@rGO (259 mV 10 mA/cm^2^), CMSO@gC_3_N_4_ (270 mV), and CMSO@SiSh (287 mV) were less than that of CMSO (384 mV), which indicated the improved OER properties of CMSO by the 2D supporting materials (Figure 5a,b). In addition, the overpotential of CMSO@rGO was less than that of RuO_2_ (315 mV). To gain insights into the OER kinetics, the Tafel slope was calculated based on overpotential and the logarithm of current density data (Figure 5c). Same as the OER overpotential, the Tafel slopes of CMSO@rGO (85 mV/dec), CMSO@gC_3_N_4_ (109 mV/dec), and CMSO@SiSh (86 mV/dec) were lower than that of CMSO (141 mV/dec), which indicated the faster OER kinetics of 2D material-supported CMSO. Among the other material-supported CMSO, CMSO@rGO exhibited the lowest Tafel slope, which was less than that of RuO_2_ (136 mV/dec).

The electrochemically active surface area (ECSA) and electrochemical impedance spectroscopy (EIS) were measured to obtain better understanding of the improved OER activity of CMSO rendered by the 2D supporting materials. The double layer capacitance (C_dl_), which was directly proportional to the ECSA value, of each material was measured from the cyclic voltammetry curves shown in Appendix A. The C_dl_ of CMSO was 2.9 mF/cm^2^, and it was increased to 11.0 mF/cm^2^ (CMSO@SiSh), 16.2 mF/cm^2^ (CMSO@gC_3_N_4_), and 35.2 mF/cm^2^(CMSO@rGO) (Figure 5d), which clearly indicated the effects of 2D materials on the C_dl_ value. The highest C_dl_ of CMSO@rGO could be strongly related to its highest BET surface area, as shown in Appendix A.

The Nyquist plots obtained from the EIS of all materials exhibited a semicircle (Figure 5e). Notably, the charge transfer resistance (R_ct_) of CMSO@rGO was measured to be 3.9 Ω, which was significantly lower than those obtained from other materials, as summarized in Table 2. This result implied that CMSO@rGO exhibited a higher electron and charge-transfer velocity than the other samples. The lower R_ct_ value of CMSO@rGO was consistent with its superior electrocatalytic activity, including lower overpotential, smaller Tafel slope, and higher ECSA. The electrochemical performances of the investigated materials are summarized in Table 2.

The role of 2D materials as templates for anchoring CMSO not only exposed more active sites to the electrolyte but also facilitated electron and charge transfer processes. The combination of the high surface area and superior conductivity of rGO might enable CMSO@rGO to achieve the fastest reaction rate compared to the other materials [51]. The observed overpotential and kinetics of CMSO@rGO fabricated herein were comparable to those of previously reported cobalt-based electrocatalysts (Appendix A and Figure 5f), revealing the high potential of CMSO@rGO as a promising electrocatalyst for OER in water electrolysis.

Stability is another key parameter to evaluate electrochemical catalysts. Chronoamperometry (CA) measurements at a constant current density were conducted to evaluate the stability of the investigated materials. The results are shown in Figure 6a. For CMSO, CMSO@rGO, CMSO@gC_3_N_4_, and CMSO@SiSh, the potentials applied to achieve a current density of ~10 mA/cm^2^ were 1.60 V, 1.49 V, 1.50 V, and 1.52 V, respectively. The current densities of CMSO, CMSO@gC_3_N_4_, and CMSO@SiSh started to decrease after 10 h of continuous operation. However, the current density of CMSO@rGO decreased negligibly, almost similar to that observed for the OER LSV curves (Figure 6b) and the unchanged morphology (Appendix A) even after 40 h of continuous operation, indicative of the superior long-term stability of as-prepared CMSO@rGO. The strong interaction between rGO sheets and CMSO might prevent the change in the morphology. Instead, a new peak was observed at 504 cm^−1^ in the Raman spectrum of CMSO@rGO after 40 h of the OER process (Appendix A), which could be attributed to the presence of CoOOH [52]. Similarly, in the XRD pattern of CMSO@rGO after the stability test (Appendix A), a new peak was observed at 20.2°, corresponding to the (003) plane of CoOOH (JCPDS No. 01-073-0497). The XPS deconvoluted spectrum of Co 2p after the stability test (Appendix A) exhibited a positive shift, and two new peaks appeared at 780 and 795 eV, respectively, originating from the Co^3+^ species in CoOOH [53]. These results revealed that the active site CMSO on the rGO sheet was partially converted into CoOOH, corresponding to the reconstruction phenomenon that occurred typically on the surface of transition-metal-based electrocatalysts in the water oxidation process [52,54]. This result suggested that the Co sites in CMSO@rGO served as favorable catalytic reaction sites for OER. The presence of a highly active CoOOH surface possibly impeded the further oxidation of the core electrocatalyst, and the interaction between the in situ oxyhydroxide and the original catalyst might be favorable in driving water oxidation. Thus, the stability of the electrocatalyst was maintained during the OER process [15,54]. The mechanism was described in the following reaction steps (* corresponds to an active site):* + OH^−^ → *OH^−^ + e^−^(1)
*OH^−^ + OH^−^ → *O + H_2_O + e^−^(2)
*O + OH^−^ → *OOH + e^−^(3)
*OOH + OH^−^ → * + O_2_ + H_2_O + e^−^(4)

A two-electrode system with CMSO@rGO as the anode and Pt/C (20%) as the cathode was designed for overall water splitting to ensure stability and scalability in large-scale industrial applications. The LSV curve (Figure 7a) of this system indicated that the potential of the cell reached 10 mA/cm^2^ at 1.54 V, while that of the RuO_2_//Pt/C system was greater by 60 mV. Even after a 40 h stability test, the chronopotentiometry curve of CMSO@rGO//Pt/C in Figure 7b exhibited an excellent activity retention of 94.8%. Particularly, the potential slightly increased from 1.54 V to 1.62 V. This result indicated that CMSO@rGO demonstrated excellent long-term stability even for the overall water-splitting reaction.

## 4. Conclusions

In this study, Co and Mo bimetallic oxide/sulfide hybrid structures supported on various 2D materials such as rGO, gC_3_N_4_, and SiSh were successfully synthesized. Among these materials, CMSO@rGO exhibited the highest electrochemical activity, with a low overpotential and a Tafel slope of 259 mV at 10 mA/cm^2^ and 85 mV/dec, respectively. Owing to the strong interaction between rGO and CMSO, the electronic structure of the composite system was modulated, promoting the formation of oxyhydroxide surfaces, and optimizing the performance of the electrocatalyst in driving water oxidation. Furthermore, the current density of CMSO@rGO changed negligibly even after a 40 h long-term stability test with no clear physical and electronic deformation, which was attributed to the high number of functional groups and high surface area of rGO.

## Data Availability

The data presented in this study are available on request from the corresponding authors.

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
