# Peer review of "Exploring the Effects of Various Two-Dimensional Supporting Materials on the Water Electrolysis of Co-Mo Sulfide/Oxide Heterostructure"

_nanomaterials, 2023, doi:10.3390/nano13172463_

Round 1

Reviewer 1 Report

In this manuscript, Huynh et al. explored the effects of various two-dimensional supporting materials, including reduced graphene oxide (rGO), graphite carbon nitride (gC3N4), and silox-ene (SiSh), on the water electrolysis (oxygen evolution reaction, OER) of CoMoO4/CoMoS hetero-structure. Of note, rGO-supported CMSO (CMSO@rGO) exhibited a large surface area and a low interface charge-transfer resistance, contributing to a better OER performance. Overall, this work has very good novelty and the results were thorough and well presented. I would like to support the publication at the journal of Nanomaterials. However, the current manuscript requires some revision to further improve the quality and clarity. The below detailed comments need to be addressed.

1. Figure 1, for the supported catalysts, the XRD peaks of SiSh and g-C3N4 were identified, while those of rGO were not seen. Why so? Please provide some explanation.

2. The format of the figures need to be revised to improve clarity. For instance, (1) Figure 3 and S9, for the SEM images, the decimal places for the scale bars are not necessary. In other words, “1.00 um” can be revised into “1 um”. (2) Figure S2-5, scale bars are missing.

3. To appeal to a broader readership, recent works about OER (Chem Eng J 2023, 10.1016/j.cej.2023.144660) are recommended to be referenced in the Introduction.

4. Figure 7, for the two-electrode overall water splitting, the term “Cell voltage (V)” is more often used compared to the term “Potential (V vs RHE)” or “Potential (V)”.

5. Table 2, for the “Overpotential (mV)” column, the authors should specific what current density this overpotential corresponds to.

6. How was the Interface charge-transfer resistance (Table 2) obtained?

7. Heterostructured materials are often considered good OER catalysts. This point should be strengthened in the Introduction. Related works (e.g., Small, 2021, 17, 2101573) can be referenced.

8. Figure 6a, why would there be fluctuation for the data of the sample CMSO?

Reviewer 2 Report

The author reported an valuable study on the 2D supporting materials for OER electrocatalysts. I recommend the publication of this study after the authors address these minor revisions.

1. The peaks in the Raman spectra should be accurately assigned.

2. The name "Co-MoO4/CoMoS (CMSO)" implies two components, but the samples actually include three phases: MoS2, CoMoO4, and CoS.

3. The spectra of pore distribution should be presented, and the authors should discuss the increase in specific surface area resulting from the integration of 2D substrates, which may not necessarily offer increased active sites.

4. Figure 3 presents two SEM images of each sample with similar resolution, which is unnecessary.

5. In the statement "Two weak peaks located at approximately 228 eV associated with S 2s," it should be noted that S2s only has one peak. The authors should clarify the meaning of "two weak peaks."

6. The authors observed a new peak at 504 cm1 in the Raman spectrum of CMSO@rGO after 40 h of the OER process, which could be attributed to the presence of CoOOH. However, the catalyst changed during the measurement, but the electrocatalytic performance remained nearly unchanged. The authors should explain the reason by citing relevant references such as Chemical Engineering Journal, 2022, 441, 136121; Chemical Engineering Journal, 2023, 455, 140821; Journal of Colloid and Interface Science, 2023, 642, 574–583; Small Struct. 2023, 2300028, doi.org/10.1002/sstr.202300028; Appl. Catal. B, 2021, 299, 120678.

7. The statement "CMSO@rGO exhibited an extremely low OER overpotential and a Tafel slope of 259 mV (10 mA/cm2) and 85 mV/dec, respectively" may not be appropriate, as several catalysts with better performance have been reported in papers such as Appl. Catal. B, 2021, 299, 120678; Adv. Sci., 2021, 8, 2101775; Appl. Surf. Sci. 2023, 619, 156789.
